# Effects of Six Consecutive Years of Irrigation and Phosphorus Fertilization on Alfalfa Yield

**DOI:** 10.3390/plants12112227

**Published:** 2023-06-05

**Authors:** Xinle Li, Jingyuan An, Xiangyang Hou

**Affiliations:** 1Experimental Center of Desert Forestry, Chinese Academy of Forestry, Dengkou 015200, China; nxylxl@126.com (X.L.); anjy326@163.com (J.A.); 2College of Grassland Science, Shanxi Agricultural University, Jinzhong 030801, China

**Keywords:** *Medicago sativa* L., dry matter yield, irrigation, fertilization, phosphorus residual effect

## Abstract

Alfalfa (*Medicago satiua* L.) is a major forage legume in semi-arid regions such as North China Plain and is the material foundation for the development of herbivorous animal husbandry. How to improve the yield of alfalfa per unit area from a technical perspective and achieve high-yield cultivation of alfalfa is the focus of research by scientific researchers and producers. To evaluate the effects of irrigation and P fertilization as well as the P residual effect on alfalfa yield, we conducted a six-year (2008–2013) field experiment in loamy sand soil. There were four irrigation levels (W0: 0 mm, W1: 25 mm, W2: 50 mm, W3: 75 mm per time, four times a year) and three P fertilization levels (F0: 0 kg P_2_O_5_ ha^−1^, F1: 52.5 kg P_2_O_5_ ha^−1^, F2: 105 kg P_2_O_5_ ha^−1^ per time, twice a year). The highest dry matter yield (DMY) was obtained in the W2F2 treatment, with an annual mean of 13,961.1 kg ha^−1^. During 2009–2013, the DMY of first and second-cut alfalfa increased significantly with increasing irrigation levels, whereas the opposite pattern was observed in fourth-cut alfalfa. Regression analysis revealed that the optimal amount of water supply (sum of seasonal irrigation and rainfall during the growing season) to obtain maximum DMY was between 725 and 755 mm. Increasing P fertilization contributed to significantly higher DMY in each cut of alfalfa during 2010–2013 but not in the first two growing seasons. The mean annual DMY of W0F2, W1F2, W2F2, and W3F2 treatments was 19.7%, 25.6%, 30.7%, and 24.1% higher than that of W0F0 treatment, respectively. When no P fertilizer was applied in F2 plots in 2013, soil availability and total P concentrations, annual alfalfa DMY, and plant nutrient contents did not differ significantly compared with those in fertilized F2 plots. Results of this study suggest that moderate irrigation with lower annual P fertilization is a more environmentally sound management practice while maintaining alfalfa productivity in the semi-arid study area.

## 1. Introduction

Alfalfa (*Medicago sativa* L.) is one of the most important forage crops throughout the world [1]. This widely adapted perennial legume has outstanding yield potential and high feeding value [2]. Like other legumes, alfalfa can fix atmospheric nitrogen (N) and produce high yields without N fertilization [3]. Given its high level of digestible protein, alfalfa is an extraordinarily valuable feed for cattle and other livestock [4,5].

China is the second largest producer of alfalfa in the world in terms of planted area, which will continuously expand with the adjustment of the agricultural production structure by the Chinese government [6,7]. The research focus on alfalfa cultivation is how to increase per unit area yield from a technical perspective [8]. Water and fertilizer are two key factors in agricultural production that affect crop growth [9,10] and can be regulated [11]. The rational and efficient use of water and fertilizer is essential for dealing with water scarcity while reducing fertilizer costs and energy consumption [12,13]. More importantly, it provides an effective way to realize the sustainable development of agriculture [14].

Alfalfa has a well-developed root system, which enables it to absorb water from deeper soil layers and enhances drought resistance [15,16]. Generally, alfalfa requires more water than other crops [17]. When its water requirement is met in a timely and adequate manner, alfalfa can be harvested five or six times a year, with dry matter yield (DMY) of as much as 15,000–25,000 kg ha^−1^ [18]. However, in the semi-arid region of North China Plain, seasonal drought has severely constrained alfalfa production. As a consequence, the DMY of alfalfa in this region is only 7500–10,000 kg ha^−1^ [19] and even lower (4500–6000 kg ha^−1^) in saline soils [20]. Additionally, the water supply for alfalfa in the growing season differs from one season to another and even from one harvest to another [21]. Current irrigation strategies are still insufficient for high alfalfa yield and efficient water use in the North China Plain.

Alfalfa yield also varies depending on soil phosphorus (P) conditions because P can affect photosynthesis, photoassimilate transportation, and plant growth [22]. P deficiency is a critical issue in North China Plain, where >75% of the agricultural land areas are deficient in P [23,24]. A large amount of P fertilizer is applied every year to mitigate P deficiency and enhance crop productivity. The P diffusion rate is influenced and determined by numerous factors (irrigation time, soil texture, P source, etc.) [25]. Up to 80% of the P applied is lost because it becomes immobile and unavailable for plant uptake due to adsorption, precipitation, and/or conversion to organic forms [26]. In this context, how much P fertilizer should be applied suitably and efficiently is a key research question that needs to be addressed for the maintenance of high alfalfa yield [27]. Additionally, the residual effect of P fertilizer on alfalfa yield is still poorly understood since previous studies mainly lasted for two or three years [28].

Long-term field experiments allow us to determine the effects of experimental treatments in a specific station under different environmental conditions [29] and as such, provide accurate and reliable information for local production [30,31]. In this study, we investigated the yield responses of alfalfa to different irrigation and P fertilization treatments over six consecutive years. Additionally, we evaluated the residual effect of P fertilizer and the relationship between annual DMY and water utilization of alfalfa. The ultimate goal was to establish proper water and P fertilizer management guidelines for alfalfa production in semi-arid regions in China.

## 2. Results

### 2.1. Effects of Year, Irrigation, and P Fertilization on Alfalfa Yield

Statistical probabilities of the *F* test for the effects of year, irrigation, P fertilization, and their interactions on the annual DMY of alfalfa are summarized in Table 1. There were highly significant differences in annual DMY between various years and between P fertilization levels (*p* < 0.01). Annual DMY was also significantly affected by the two-way interaction of irrigation × P fertilization, year × irrigation, and year × P fertilization (*p* < 0.05), but not affected by different irrigation levels or the three-way interaction of year × irrigation × P fertilization.

### 2.2. Effects of Irrigation on Yield of Different Cut Alfalfa

Irrigation significantly improved the DMY of the first- and second-cut alfalfa in most cases, excluding 2008 and the second cut of 2013 (Figure 1). The opposite effect was observed for the DMY of fourth-cut alfalfa in different years, excluding 2008 and 2013. The DMY of first and second-cut alfalfa increased with increasing irrigation levels (W3 > W2 > W1 > W0), in contrast to the pattern of fourth-cut alfalfa (W0 > W1 > W2 > W3). The DMY of first, second, and fourth cut alfalfa showed consistent variation in response to irrigation over the six years. However, the effect of increasing irrigation on the DMY of third-cut alfalfa varied across different years, leading to considerable yield improvement in 2009 and 2013, different yield decline in 2008, 2011, and 2012, and no yield change in 2010.

### 2.3. Effects of P Fertilization on Yield of Different Cut Alfalfa

The effects of P fertilization on alfalfa DMY were significant for each cut from 2010 to 2013, despite no significant differences in DMY between various P fertilization levels in the previous two years (Figure 2). Increasing P fertilization did not improve DMY in 2008 and 2009 but exhibited a positive effect on DMY in the following four years. Compared with the F0 treatment, the increase of annual DMY in the F1 and F2 treatments reached 1305 and 1964 kg ha^−1^ in 2010, 2176 and 4194 kg ha^−1^ in 2011, 2933 and 4858 kg ha^−1^ in 2012, and 1334 and 2298 kg ha^−1^ in 2013, respectively. The six-year mean DMY under P fertilization increased by 1.05–29.05% and 2.96–48.11% in the F1 and F2 treatments, respectively.

In 2013, there were no significant differences between F2 and F2′ treatments in terms of soil available and total P concentrations, as well as annual alfalfa DMY and major nutrient contents (Figure 3). The DMY of F2′ treatment with no P fertilization accounted for 92.2–103.6% of the DMY of F2 treatment. This indicates that the P fertilizer applied in previous years had a residual effect. Fertilizer P that accumulated in the soil could satisfy the requirement of alfalfa crops.

### 2.4. Coupling Effects of Irrigation and P Fertilization on Alfalfa Yield

The six-year mean DMY of alfalfa in different treatment combinations followed the order of W2F2 > W2F1 > W1F2 > W3F2 > W1F1 > W3F1 > W2F0 > W0F2 > W0F1 > W3F0 > W1 F0 > W0F0 (Table 2). Due to insufficient water and fertilizer supply, the annual mean DMY of the control treatment (W0F0) was the lowest of all, only 10,681.8 kg ha^−1^. The DMY of other treatment combinations variably increased by 12.5–30.7% compared with that of the W0F0 treatment. Moreover, under the same level of P fertilization, the DMY in irrigated plots was significantly higher than that in non-irrigated plots. Accordingly, under the same level of irrigation, the DMY in plots with high-level P fertilization (W0F2, W1F2, W2F2, W3F2) was higher than that in other plots, and their yield increase relative to the W0F0 treatment was 19.7%, 25.6%, 30.7%, and 24.1%, respectively. The highest DMY was observed in the W2F2 treatment, with an annual mean of 13,961.1 kg ha^−1^. In addition, based on the measured soil and alfalfa P content data, it was found that the W2F2 treatment with total P content of 0.92 g/kg and 0.41 g/kg in the 0–20 cm and 20–40 cm soil layer, available P content of 18.7 mg/kg and 4.6 mg/kg, and total P content in alfalfa is 4.2 g/kg. These indicators were the highest in all experimental treatments.

### 2.5. Relationship between Alfalfa Yield and Water Supply

During the 2008–2013 growing seasons, the climatic conditions (especially rainfall) at the experimental site were highly variable, resulting in significant DMY differences across the years. To quantitatively describe the relationship between the total amount of water supply (seasonal irrigation amount + growing season rainfall) and the annual DMY of alfalfa, regression analysis was carried out to establish regression equations. The results consistently demonstrated a significant correlation between annual DMY and water supply in each year (*p* < 0.01; Figure 4). Annual DMY linearly increased with an increasing amount of water supply in 2009 and 2010 (dry seasons). However, the relationship between annual DMY and water supply was described by a quadratic curve in 2008, 2011, and 2012 (wet seasons), as well as in 2013 (normal season). This means that with an increasing amount of water supply, annual DMY initially increased to a peak and then declined in the wet and normal seasons. Further analysis revealed that the optimal amount of water supply (optimal water requirement) for alfalfa to obtain the maximum DMY was between 725 and 755 mm under the experimental conditions.

## 3. Discussion

### 3.1. Alfalfa Responses to Irrigation and Optimal Water Supply

Lack of available water has become the primary factor limiting crop yields in the North China Plain [32], where only appropriate irrigation can mitigate the adverse effects of water deficit [33,34]. In the semi-arid study area, first and second-cut alfalfa was grown in the relatively dry season, and the mean rainfall received in these two periods was only 47.6 and 61.1 mm, respectively. Owing to the absence of adequate rainfall and considerable soil water consumption by transpiration for dry matter synthesis [35], the water expenditure was far greater than the water income of first and second-cut alfalfa [36], resulting in a serious water deficit.

Previously, Saeed et al. [37] found that water deficit reduced the stem height and density, leaf area index, total biomass production, and water use efficiency of alfalfa plants. Brown et al. [38] reported that mild drought in the early growth stage of alfalfa caused a yield reduction of 15% because of lower stem density. Water deficit-induced reduction in alfalfa transpiration is associated with a decrease in biomass production [39,40]. Since the water was the most important factor in limiting alfalfa growth in the first and second cuts, supplemental irrigation strongly promoted plant growth and improved crop yield over six consecutive years.

Fourth-cut alfalfa was grown in the rainy season, and the lowest amount of rainfall received in this period reached 195.2 mm in 2010. Consequently, water was no longer a limiting factor for alfalfa, and excessive rainfall even caused waterlogging. The susceptibility of alfalfa to waterlogging injury seriously could limit its persistence and adaptability, resulting in substantial yield losses [41]. Halim et al. [42] found that when flooded for 7 days, alfalfa exhibited severe shoot injury symptoms (wilted and yellowing leaves), with reduced shoot dry weight and decreased storage of total nonstructural carbohydrates in shoots and roots. These findings demonstrate why increasing irrigation negatively affected alfalfa yield in the fourth cut. Accordingly, there was no need to irrigate during the growth period of fourth-cut alfalfa.

A myriad of studies has established a linear relationship between annual DMY and water supply for alfalfa [43,44]. However, this relationship may vary across climatic regions [45], seasons [46], and even cuttings within the growing season [47,48]. Our results showed that the annual DMY of alfalfa had a linear relationship with the amount of water supply in the dry seasons (2009, 2010) and a quadratic relationship in the wet seasons (2008, 2011, 2012). Furthermore, we optimized the irrigation regime for alfalfa by considering its water supply, annual rainfall, and the effects of irrigation on the DMY of four cuts. Irrespective of the dry or wet season, the growth of first- and second-cut alfalfa required extensive irrigation (W3). The situation of third-cut alfalfa was different, which required maintaining moderate irrigation (W2) in the dry seasons and stopping irrigation in the normal and wet seasons. Fourth-cut alfalfa had no need for irrigation under experimental conditions.

### 3.2. Alfalfa Responses to P Fertilization and P Residual Effect

While essential for crops, P can be regarded as a growth-limiting nutrient because its availability to plants is relatively low due to poor solubility and high immobilization rate in soil [49,50] Excluding the first two years of the experiment, P fertilization increased alfalfa DMY during both the wet and dry seasons. This study’s result is partially different from the finding of Berg et al. [51], which showed that incremental P fertilization P improved alfalfa yield each year. Unabsorbed P became a residue in the soil, which could exist in different forms and accumulate over time [52]. Additionally, alfalfa DMY in plots receiving high-level P fertilization was higher than in plots receiving low-level P fertilization. However, excessive P fertilization is undesirable because of potential environmental risks, such as water pollution. Since the annual rainfall was different, alfalfa responded deferentially to P fertilization in the dry and wet seasons. Across the six years, alfalfa DMY in the dry seasons (2009, 2010) was 10.7–15.4% lower than in the wet seasons (2008, 2011, 2012) under P fertilization.

In this experiment, the main component of P fertilizer is monocalcium P. When its particles enter the soil, they absorb water from the soil, forming a saturated solution containing phosphoric acid and dicalcium phosphate. This strongly acidic saturated solution begins to diffuse outside the fertilizer particles and is then absorbed by alfalfa. Phosphorus dissolved in the soil solution is adsorbed into the soil by the soil particles. The greater the binding capacity of the adsorbed phosphorus, the smaller the fertilizer effect. However, the adsorbed phosphorus can be further converted and absorbed by alfalfa over time.

When P fertilization was stopped, P fertilizer still had a remarkable residual effect on alfalfa yield in the last year of the experiment. For example, high-level P fertilization (F2) increased annual DMY to approximately 12,000–16,000 kg ha^−1^. This yield target was maintained after stopping P fertilization through the residual effect of previously applied P fertilizer. Long-term experiments have demonstrated that superphosphate is a slow-release fertilizer [53,54], and as such, residual P in the soil could be slowly released for crop uptake and utilization in the subsequent years [55]. By comparing the effects of one-time initial and annual P applications, Moyer [56] found that P fertilizer was available to perennial forages for 4–10 years in relatively dry regions of the Canadian prairies. According to Xi [57], the apparent P fertilizer use efficiency (residual effect of P counted) can reach 36–44%. Furthermore, Tian et al. [58] observed the superimposed residual effect of P in a four-year experiment, which strongly improved the seasonal fertilizer yield increase rate.

In summary, when P fertilizer is applied to the soil, only a portion of P fertilizer can be utilized by alfalfa in the first two seasons. The unused and immobilized P is initially unavailable to plants but will be transformed into available forms and slowly absorbed by crops in later years. For alfalfa production, it is necessary to consider both the yield-increasing effect of P fertilizer and the effect of residual P in the soil for rational and economical fertilization. Furthermore, the plant growth responses and root plasticity of alfalfa with P fertilization under various soil moisture stresses should be exploited and manipulated to enhance its yield production in water-stress/-scarce environments. The mechanisms underlying the residual effect of P fertilizer, including the turnover of P recycled from alfalfa plants and long-term reactions of P fractions in sandy soil, also need to be investigated [59].

### 3.3. Alfalfa Responses to the Coupling of Irrigation and P Fertilization

The coupling of water and fertilizer emphasizes the need to take advantage of the synergy between two factors for integrated management of water and fertilizer in order to increase crop productivity as well as water and fertilizer use efficiency [60]. In this study, irrigation and P fertilization showed significant interaction effects on the annual DMY of alfalfa. Under the same level of P fertilization (F2), irrigation treatments (W1F2, W2F2, W3F2) prominently increased alfalfa DMY compared with no irrigation treatment (W0F2). Similarly, under the same level of irrigation, P fertilization treatments (W1F1, W1F2, W2F1, W2F2, W3F1, W3F2) contributed to higher alfalfa DMY compared with no fertilization treatments (W1F0, W2F0, W3F0).

Myriad greenhouse and field studies have looked at the relationship between water and fertilizer. Evidence suggests that coordination of irrigation and fertilization can improve seedling development as well as increase root biomass and leaf area index [61,62]. Our results mirrored previous findings of Wang et al. [63] in that irrigation combined with P fertilization improved the DMY of alfalfa plants, which exhibited delayed browning in autumn and earlier greening in spring. Additionally, Helalia et al. [64] observed a pronounced interaction effect between water quality and P fertilizer on increasing the DMY of alfalfa over two growing seasons. The collective results suggest that appropriate water supply can facilitate P transformation and absorption by alfalfa plants, improving their fertilizer use efficiency. Proper fertilization can also regulate water use by alfalfa plants and thereby enhance their water use efficiency.

## 4. Materials and Methods

### 4.1. Experimental Site

The field experiment was conducted at Langfang Experiment Station (116°34′60′′–116°36′13′′ E, 39°35′44′′–39°36′14′′ N), Chinese Academy of Agricultural Sciences. It is located in the northern part of Langfang (Hebei Province, northern China)—a major region for farming and aquaculture in Beijing. The experimental site has an elevation of 25 m above sea level, and it belongs to a temperate semi-arid and semi-humid continental climate zone. The mean annual rainfall is 554.9 mm, 70% of which occurs in the monsoon season (June–August). Spring drought is often a limiting factor for seed germination, regreening, and growth of alfalfa. The original soil at the experimental site was loamy sand, and the topsoil (0–20 cm) contained 16.9% of clay content, 11.0 g kg^–1^ of organic matter, 0.9 g kg^–1^ of total P, 6.2 mg kg^–1^ of available P, 0.7 g kg^–1^ of total N, and 42 mg kg^–1^ of available N, as determined by the method of Bao [65]. The soil pH measured with a soil: water ratio of 1:2.5 *w*/*v* was 8.52.

From 2008 to 2013, monthly rainfall and temperature (Figure 5) were measured at a meteorological station located ~50 m away from the experimental site. Most rainfall occurred between June and September. Additionally, the growth period rainfall was recorded for each cut of alfalfa (Table 3), which indicates that 2008, 2011, and 2012 were wet seasons, 2009 and 2010 were dry seasons, and 2013 was a normal season.

### 4.2. Experimental Design and Treatments

The experiment commenced in 2008 and ended in 2013, comprising one normal, two dry, and three wet seasons. It adopted a factorial randomized complete block design with 12 treatments (4 × 3) in three replicates. The first factor included four levels of irrigation: no irrigation (W0) and irrigation at 25 mm (W1), 50 mm (W2), and 75 mm (W3) per time. Plots were irrigated four times a year after the regreening stage and the first, second, and third cuts, respectively. The second factor included three levels of P fertilization: no P (F0), 52.5 kg P_2_O_5_ ha^−1^ (F1), and 105 kg P_2_O_5_ ha^−1^ (F2) per time, The amount of P fertilizer used in reference to Wen’s research [66]. P fertilizer was applied as superphosphate, namely Ca(H_2_PO_4_)_2_, (P_2_O_5_ = 12%, Sulfur = 10%) twice a year, after the regreening stage and the second cuts, respectively. P fertilizer is manually applied to the soil surface of the experimental community. Subplots (6 m × 6 m) were separated with double bunds that were spaced 50 cm apart to prevent water flow between plots.

In 2013, the experimental design was adjusted on the basis of the results from 2008–2013 to determine the residual effect of P fertilizer. F2 plots were divided into two parts each: one part still received the same amount of P fertilizer (F2: 210 kg P_2_O_5_ ha^−1^), and the other part received no P fertilizer (F2′: 0 kg P_2_O_5_ ha^−1^).

### 4.3. Crop Management and Yield Measurement

Medicago sativa L. Zhongmu 1 characterized by high yield potential and tolerance to a large range of pests and diseases, was used in the experiment. Alfalfa seeds were broadcasted in flooded plots at a density of 22.5 kg seeds ha^−1^ and a depth of ~2 cm on 1 October 2007. The field was surface irrigated immediately after sowing. Seed germination and stand establishment were recorded. There were four cuttings per year, from budding to early flowering. At each cutting, three quadrats (1 m × 1 m) were chosen randomly in each plot, and plants were cut at a height of 5 cm above the ground level. Then, 200 g subsamples were deactivated at 105 °C for 15 min, oven-dried at 70 °C for 48 h, and weighed to determine DMY. After that, all plants in each plot were mowed.

In 2013, plant samples (including stems and leaves) of F2 and F2′ treatments were collected at the fourth cutting to analyze N, P, and potassium (K) contents. The determination of N, P, and K content in alfalfa plants is carried out using conventional soil agrochemical analysis methods [65]. After drying and crushing the plant samples in each experimental community, their total N content was determined using the Kjeldahl method, total P content was determined using molybdenum blue colorimetry, and total K content was determined using flame photometry.

One week after harvest, soil samples were collected from depths of 0–20 and 20–40 cm to determine soil available and total P concentrations under the two different treatments.

### 4.4. Statistical Analysis

For each cropping season, a two-way analysis of variance (ANOVA) was carried out to determine the treatment effects on alfalfa yield using SAS v9.1.3 (SAS Institute Inc., Cary, NC, USA). When treatment effects were significant, group means were compared using the least significant difference (LSD) test at the 0.05 level. The relationship between alfalfa yield (annual DMY; kg ha^−1^) and water supply (sum of irrigation and rainfall; mm) was assessed using a general-purpose procedure (PROC REG) for regression analysis.

## 5. Conclusions

Through a six-year field experiment, we demonstrated the coupling effects of irrigation and P fertilization on increasing alfalfa yield in the semi-arid region of North China Plain. Irrigation at 50 mm per time (four times a year) combined with P fertilization at 105 kg ha^−1^ per time (twice a year) resulted in the highest annual DMY of alfalfa (13,961.1 kg ha^−1^). Annual DMY was strongly influenced by the amount of water supply during the growing season, based on a linear relationship (dry seasons) or a quadratic relationship (wet seasons). The optimal amount of water supply for alfalfa to obtain its maximum DMY was between 725 and 755 mm (sum of seasonal irrigation and rainfall). The application of 210 kg P_2_O_5_ ha^−1^ enabled annual DMY to reach 12,000–16,000 kg ha^−1^. This yield target was maintained after stopping P fertilization for one year owing to the residual effect of previously applied P fertilizer. In addition, an application level of 210 kg P_2_O_5_ ha^−1^ can effectively improve the phosphorus content in the soil and ensure the effective absorption of phosphorus by alfalfa. The findings of this study can be useful for farmers to develop a balanced irrigation and P fertilization regime for alfalfa according to annual rainfall, crop water requirement, and P residual effect. Application of such a balanced regime can reduce the need for a greater amount of water and fertilizer without compromising crop yield.

## Figures and Tables

**Figure 1 plants-12-02227-f001:**
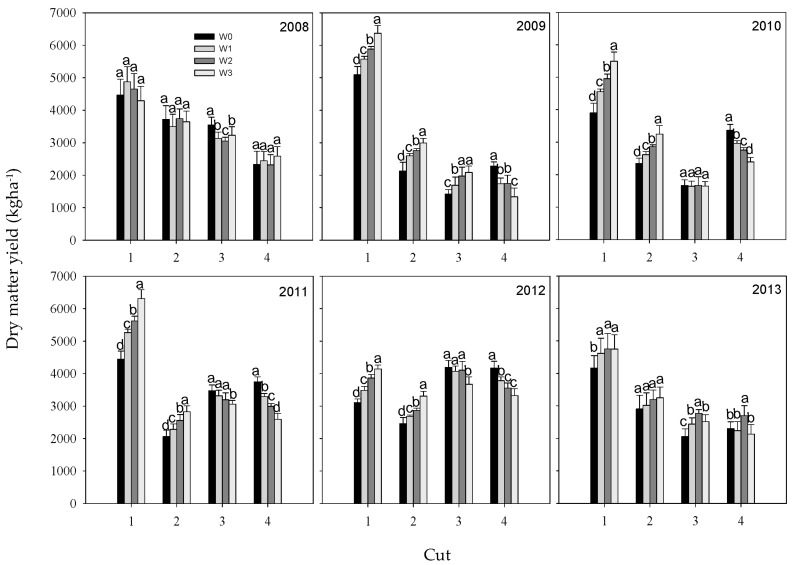
Dry matter yield of different cut alfalfa under four irrigation levels in six consecutive years. Values with different lowercase letters in the same column were significantly different at the 0.05 level.

**Figure 2 plants-12-02227-f002:**
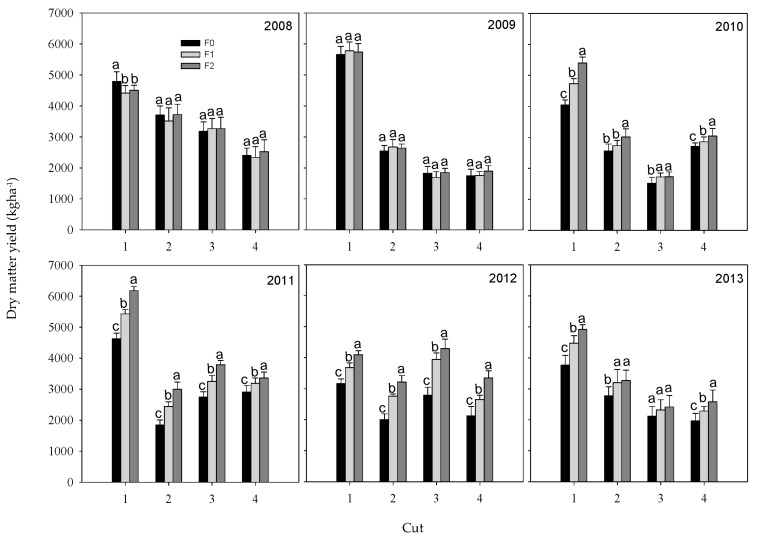
Dry matter yield of different cut alfalfa under three P fertilization levels in six consecutive years. Values with different lowercase letters in the same column were significantly different at the 0.05 level.

**Figure 3 plants-12-02227-f003:**
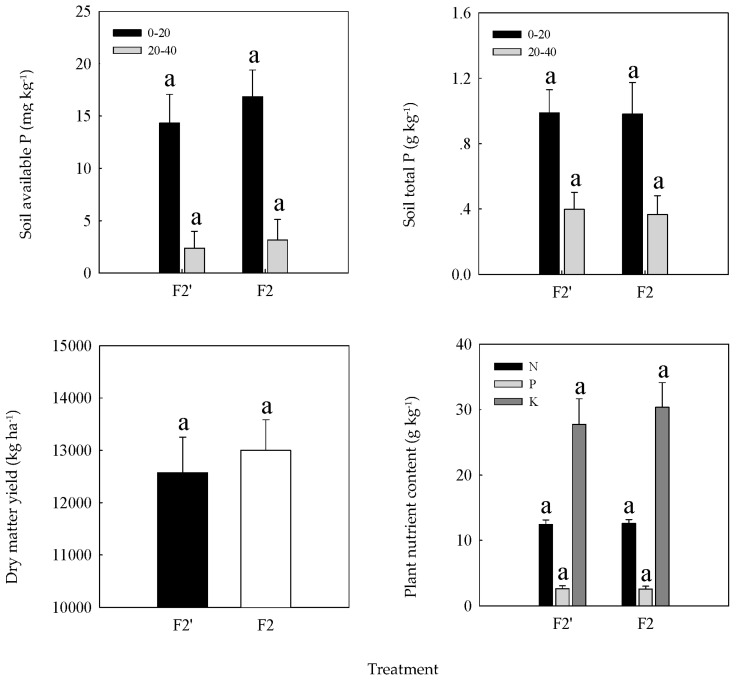
Differences in soil P concentration, alfalfa yield, and major nutrient contents between F2 (with P fertilization) and F2′ (with no P fertilization) treatments in 2013. Values with different lowercase letters in the same column were significantly different at the 0.05 level.

**Figure 4 plants-12-02227-f004:**
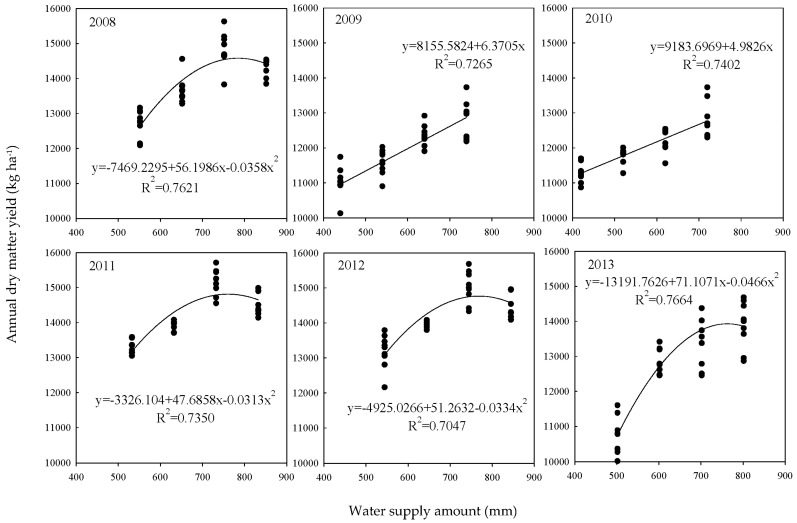
Relationship between annual alfalfa yield and growing season water supply.

**Figure 5 plants-12-02227-f005:**
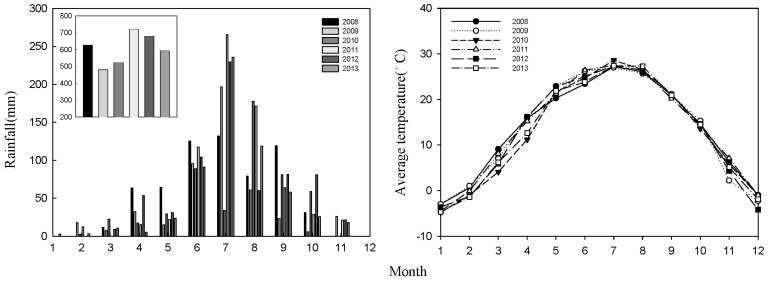
Monthly rainfall and temperature at the experimental site (2008–2013).

**Table 1 plants-12-02227-t001:** Statistical probabilities of *F* test for year, irrigation, P fertilization, and their interactions on annual dry matter yield of alfalfa.

Source	df	*F* Value	Significance
Year (Y)	5	110.25	**
Irrigation (W)	3	2.39	ns
P fertilization (F)	2	10.28	**
Y × W	15	1.87	*
Y × F	10	2.57	*
W× F	6	0.73	*
Y × W × F	30	0.89	ns

Note: *, *p* < 0.05; **, *p* < 0.01; and ns, not significant at the 0.05 level.

**Table 2 plants-12-02227-t002:** Annual yield of alfalfa in different irrigation and P fertilization treatments (kg ha^−1^).

Treatment	2008	2009	2010	2011	2012	2013	Mean
W0F0	14,276.9 ^a^	11,245.9 ^a^	10,786.5 ^c^	12,582.5 ^c^	12,156.5 ^c^	9153.8 ^g^	10,681.8
W0F1	13,638.1 ^a^	11,609.1 ^a^	11,677.6 ^bc^	14,402.5 ^bc^	14,436 ^ab^	11,544.6 ^e^	12,348.7
W0F2	14,271.3 ^a^	11,740.3 ^a^	11,890.5 ^ab^	15,110.8 ^ab^	15,143.4 ^ab^	11,933.4 ^e^	12,782.4
W1F0	14,184.9 ^a^	11,587.9 ^a^	11,215 ^bc^	13,279.1 ^bc^	13,510.6 ^bc^	11,819.4 ^e^	12,287.5
W1F1	13,561.9 ^a^	12,082.9 ^a^	12,359.5 ^ab^	14,860.6 ^ab^	14,710.3 ^ab^	12,617.9 ^d^	13,066.5
W1F2	14,099.2 ^a^	12,191.3 ^a^	12,385.3 ^ab^	14,862.5 ^ab^	15,332.6 ^ab^	13,068.1 ^c^	13,421.2
W2F0	13,550.5 ^a^	12,431.3 ^a^	11,891.5 ^ab^	13,817.5 ^bc^	13,965 ^bc^	12,599.8 ^d^	12,865.5
W2F1	13,784.7 ^a^	12,510.0 ^a^	12,747.6 ^ab^	14,478.9 ^ab^	15,048.4 ^ab^	13,558.3 ^b^	13,636.1
W2F2	13,967.3 ^a^	12,253.2 ^a^	12,958.4 ^a^	15,712.5 ^a^	15,686.8 ^a^	13,806.5 ^a^	13,961.1
W3F0	14,321.3 ^a^	11,872.4 ^a^	11,632.2 ^bc^	13,685.2 ^bc^	13,934.2 ^bc^	10,952.2 ^f^	12,020.7
W3F1	13,168.2 ^a^	12,289.2 ^a^	12,226.9 ^bc^	14,792.4 ^ab^	13,963.6 ^bc^	12,568.3 ^d^	12,928.2
W3F2	13,792.7 ^a^	12,147.5 ^a^	12,514.2 ^ab^	14,567.2 ^ab^	13,546.6 ^bc^	13,197.6 ^b^	13,255.6

Note: Values with different lowercase letters in the same column were significantly different at the 0.05 level. W0 to W3 indicate no irrigation and irrigation at 25, 50, and 75 mm per time, respectively (four times a year). F0 to F2 indicate no P and P fertilization at 52.5 and 105 kg P_2_O_5_ ha−1 per time, respectively (twice a year).

**Table 3 plants-12-02227-t003:** Growth period rainfall (mm) for each cut of alfalfa at the experimental site.

Cut	Year						
	2008	2009	2010	2011	2012	2013	Mean
First	88.6	35.0	42.3	40.7	57.0	22.0	47.6
Second	96.1	56.0	49.2	58.4	60.0	46.6	61.0
Third	153.6	105.0	133.9	132.4	149.6	199.9	145.7
Forth	213.4	244.0	195.2	301.0	277.8	233.3	244.1
Growing season	551.7	440.0	420.6	532.5	544.4	501.8	498.5

## Data Availability

Not applicable.

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
