# Peer review of "Effects of Six Consecutive Years of Irrigation and Phosphorus Fertilization on Alfalfa Yield"

_plants, 2023, doi:10.3390/plants12112227_

Round 1

Reviewer 1 Report

This manuscript was interesting and meaningful. My major concerns were as followings:

1. Abstracts section could be improved to highlight the theme of this study. Significant results were necessary, not only the percentages.

2. In the introduction section, recent references could be added and meaning of this study should be focused.

3. How about the experimental replication? Please to be clearly stated in M&M section.

4. Please focus on the data with significant differences in the results section.

5. In the discussion section, please comparatively discuss the main results with previous references. Some results could be concise.

English Language could be improved by native English Speakers.

Reviewer 2 Report

The present research investigated the effect of P application and irrigation levels on alfalfa production over six years. Thus, the research is of great importance to improve the practice of fertilization and irrigation of alfalfa cultivation in China.

P can be very adsorbed in the soil depending on the clay content and the pH value, among other factors, and this decreases the P diffusion rate in the soil and consequently the nutrient uptake by the plant and the crop response. The diffusion rate of P in the soil as a function of the water content in the soil needs to be more detailed in the introduction and discussion of the work.

In this type of experimentation, it is important for the authors to establish the critical levels of P in the soil and P in the leaf for the treatment that resulted in the highest productivity. This research can suggest these critical contents in the soil and in the leaf, as it can contribute to the evaluation of the nutritional status of alfalfa using these P contents as a reference. This needs to be discussed in the manuscript. There are other questions that I leave indicated below.

- Indicate the clay content of the soil;

- Indicate the full name of the P source and the P2O5 content. If this source has sulfur in its composition, I ask if this was balanced in the different doses of phosphorus used in the research.

- It lacked details on how to apply phosphorus in cultivation. It was distributed on the soil surface or applied in a row and incorporated into the soil. The fertilizer was applied manually or by machine.

- And the other nutrients as they were applied in the experiment.

- Indicate the reference used to define the doses of phosphorus in the experiment;

- It was indicated that the plant was collected for chemical analysis, but it lacked detailing which leaf was collected and a reference that recommends that type of leaf.
